# AN APPROACH FOR LARGE LANGUAGE MODEL DEPLOYMENT IN DISTRIBUTED EDGE ENVIRONMENTS

## ABSTRACT

Large Language Models (LLMs) have made a huge impact on tasks like question answering and text generation. However, they still struggle with consistency and efficiency, especially when deployed on resource-limited devices. In this paper, we explore using multiple LLMs to work together in a multi-agent system to improve performance. We use two large language models to generate multiple answers, and then combine the answers using another large language model. We also look at how reducing the precision of these models using quantization affects performance, comparing full-precision and INT8 versions. Our experiments, using questions from the Natural Questions dataset, show that combining smaller models results in more accurate and stable answers than relying on just one model. Quantization helps make the models more efficient, without sacrificing too much performance, making them easier to deploy on devices with limited computational power.

## 1 INTRODUCTION

Large Language Models (LLMs) have greatly transformed the field of language-based tasks, especially in areas like text generation, answering questions, and language understanding. One of the main reasons for the success is the transformer architecture, introduced by Vaswani et al. The Transformer uses a technique called self-attention, which allows the model to process information all at once, instead of one piece at a time, i.e., parallel processing of information. This ability to handle large amounts of data at once makes LLMs much more efficient and capable of generating meaningful context-sensitive responses.

Despite the success of LLMs, they still face some challenges Kaddour et al. (2023). One significant issue is the consistency of the model. Although LLMs generally provide accurate responses, their responses may vary, especially when the same question is asked in different ways. Small changes in input can result in very different outputs. This becomes a problem in real-world situations where consistency, clarity are critical, such as in fields like healthcare, law, and finance Lu et al. (2024). Also, larger models use more computational resources, making them difficult to scale and deploy, especially in environments with limited resources like edge devices Dong et al. (2023).

To address these challenges, our work has explored multi-agent systems, where multiple LLMs collaborate rather than compete Cemri et al. (2025). In this framework, independent models provide different answers to a task, and a secondary model synthesizes these answers into a unified output bearing the context. This approach leverages the strengths of each model, improving both the accuracy and reliability of the system Lu et al. (2024); LangChain AI (2024). By combining multiple LLMs, the system can also erase the inconsistencies of individual models. Thus providing us with more accurate and stable results. This work aligns with ensemble machine learning approaches Ganaie et al. (2022) but extends the investigation to large language models (LLMs). The primary objective is to explore the adaptability of LLMs in similar settings while enabling a distributed architecture capable of operating on edge devices. In our formulation, each LLM can be replaced by an individual Retrieval-Augmented Generation (RAG) model, which has access to domain-relevant data for inference. Although RAGs are often employed for knowledge-insensitive NLP tasks, we propose the use of smaller, domain-specific RAG models whose retrieved knowledge

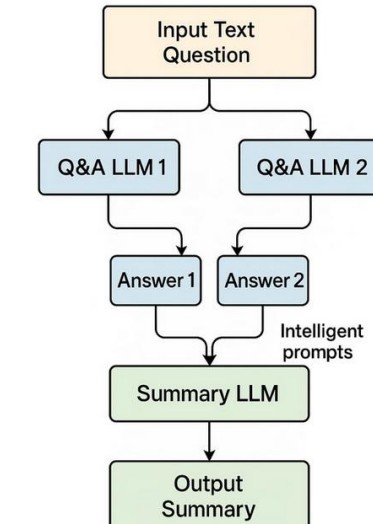

Figure 1: A multi-agent LLM system where answers from two QA models are combined by a summarization model to generate a final response.

can be aggregated. This approach represents an additional contribution of our work Lewis et al. (2020). Moreover, the proposed framework offers potential benefits for explainable AI. Specifically, it enhances the ability to trace and identify the underlying models responsible for providing domain-specific information Mersha et al. (2024).

In addition to multi-agent systems, model quantization offers a way to make LLMs more scalable. Reducing the precision of the parameters of a model, for example, from 16-bit floating point to 8-bit integer quantization, reduces the size of the model, making it easier to deploy on edge devices with limited resources, maintaining competitive performance Reimers & Gurevych (2019); Egashira et al. (2024). In this work, we examine how quantization impacts both standalone models and collaborative models as in Figure 1.

In this paper, we propose a multi-agent framework for question answering (QA), where two independent LLMs `Pythia-1B` and `Phi-2` are used to generate answers to the same input question. Here, both the LLMs generate different answers but with the same context, to the same question. These responses are then summarized by a third model, `Llama-3.2-1B Instruct`, which combines the different responses into one output bearing the context. We also evaluate the impact of quantization on the performance of these models. Specifically, we compare the results of full-precision and INT8-quantized versions of the models on metrics like BERTScore, BLEU score, and ROUGE score. Our experimental evaluation includes 1K sample questions from the dataset called Natural Questions Kwiatkowski et al. (2019). The paper is organized as follows: Section 2 reviews related work on multi-agent systems, collaborative strategies in LLMs, and the impact of quantization. Section 3 describes the methodology behind the multi-agent QA system. Section 4 discusses the experimental setup, including the dataset, evaluation metrics, and model configurations. Section 5 presents the results of our experiments, along with the analysis of the impact of quantization on model performance. Finally, Section 6 concludes the paper, summarizing our findings and discussing future research directions.

CONTRIBUTIONS

This study addresses the following research questions as its primary contributions:

**Q1** Is the multi-agent system approach universally beneficial across various applications of large language models (LLMs), and what are its associated advantages and limitations?

**Q2** Do larger models consistently outperform smaller ones, or can combinations of quantized models achieve superior performance compared to individual large models?

**Q3** Do combinations of quantized and base models often yield similar overall performance, and what factors contribute to this outcome?

**Q4** Why is there a reduction in tokens-per-second throughput in quantized models, and what are the possible explanations for this phenomenon?

## 2 RELATED WORKS

There has been considerable recent interest in using language models for text-based tasks such as question answering and summarization. A number of studies have also explored the idea of combining multiple models for these tasks. While our approach is related, it is simpler and more focused. We use the Pythia model family Biderman et al. (2023) as one of the components in our setup. That work primarily examined how model behavior varies with scale, rather than how models can be used collaboratively. In contrast, we explore how smaller models like `Pythia-1B` and `Phi-2` can contribute together to the same task. Other recent studies LangChain AI (2024); Guo et al. (2024) have proposed multi-agent frameworks in which individual agents, i.e, LLMs, operate as autonomous agents, each assigned distinct roles. These agents may engage in follow-up questioning, verify the outputs of other models, or determine the next steps in a reasoning chain. While such systems can be powerful, they are often complex, involving multiple stages of interaction and coordination. In contrast, our setup is intentionally straightforward: two models generate responses independently, and a third model synthesizes these into a final summary. This fixed structure avoids inter-model communication and reduces system complexity, making the approach more interpretable and easier to implement. Some research, such as Lu et al. (2024), has proposed tighter integration strategies like averaging outputs or combining predictions. We do not follow that route. Instead, we treat each model's output as a separate input and combine them only at the final stage through summarization. This keeps the system easy to interpret and manage. We also consider model compression, particularly 8-bit quantization, which is important for running models efficiently on limited hardware. Prior work Dettmers et al. (2022); Dong et al. (2023) has shown that performance remains strong even after quantization. Building on this, we show that compressed models can still complement each other effectively, and that the summarization step can help offset individual model limitations. For evaluation, we use the Natural Questions dataset Kwiatkowski et al. (2019), applying standard metrics such as ROUGE and BERTScore Hu & Zhou (2024). While these are commonly used, we apply them to the final summarized output rather than to the individual model responses, giving a more realistic measure of end-to-end answer quality. In summary, our work combines small, local models in a fixed structure without complex coordination or fusion. It also examines how well this structure performs under quantization, offering a lightweight yet effective alternative to more elaborate systems. Importantly, our experiments show that combining responses from multiple small models and summarizing them leads to better performance than using any of the individual models alone, something not widely emphasized in prior work.

## 3 METHODOLOGY

In this research, we explored whether combining the outputs of multiple language models could lead to better question answering performance. The main idea was: instead of relying on a single model to answer, we used two different models `Phi-2` and `Pythia-1B` to each generate their responses to the same question. These models understand the questions and provide relevant answers, but they may differ in wording, detail, or phrasing. However, they generally maintain the same overall context and major information.

After collecting the two answers from these models, we passed them to a third model, `LLaMA-3.2-1B-Instruct`, which is a summarizer. The role of this model was to look at both responses and intelligently merge them into a single, final answer. To improve the quality of this summary, we used intelligent prompts. Thus the model extracts the most important parts of each input and phrases them clearly and concisely. These prompts helped the summarizer focus on relevance, avoid redundancy as shown in Figure 1.

To evaluate how well this system performed, we tested it using a set of 1K real-world questions taken from the Natural Questions dataset.

Our evaluation was done in three stages to get a complete picture of how the models performed under different conditions:

- *Individual testing:* First, we evaluated each model separately, running them in their original base form (full precision). This helped us understand their standalone capabilities.

- *Quantized testing:* Next, we reduced the size of each model by quantizing it to INT8 format, which makes them more efficient, especially useful for deploying on devices with limited computing power. We tested their performance again in this form.

- *Combined testing:* Finally, we ran the entire setup starting from the user question, generating two model responses and then summarizing them both in full precision and the quantized version and the quantized version.

## 4 EXPERIMENTS

### 4.1 DATASET

For our experiments, we used a sample of 1K question-answer pairs from the Natural Questions dataset, which is based on Google's Natural Questions Kwiatkowski et al. (2019). The full training set contains around 100K entries, but we selected a smaller subset. Each entry in the dataset has a real user question along with a detailed answer taken from Wikipedia. These answers often go beyond short facts and include rich, contextual information. This makes the dataset a good fit for tasks like summarization or semantic evaluation, where understanding the meaning of the text matters more than just matching words.

### 4.2 MODELS

We used three pretrained language models in our experiments: `Phi-2`, `Pythia-1B`, and `LLaMA-3.2-1B-Instruct`. `Phi-2` and `Pythia-1B` were used as question-answering agents, while `LLaMA-3.2-1B-Instruct` acted as the summarization model responsible for combining and refining their responses. Initially, all models were run in full precision, i.e, `Phi-2` and `Pythia-1B`, in FP16, and `LLaMA-3.2-1B-Instruct` in BF16 due to its architecture. We also quantized all three models to INT8 Dettmers et al. (2022) using `bitsandbytes` Belkada & Dettmers (2022). This allowed us to directly compare base and quantized variants while keeping the architecture and model weights structurally aligned. `Phi-2` is a 2.7B parameter transformer developed by Microsoft. It was trained on 1.4 trillion tokens, sourced from a combination of synthetic NLP data and filtered web content, with a focus on safety and educational relevance. It performs strongly on benchmarks involving reasoning and language understanding, and is well-suited for QA tasks. `Pythia-1B`, developed by EleutherAI Biderman et al. (2023), is a 1B parameter model from a suite built for controlled research on language models. It uses the GPT-NeoX framework and provides access to training checkpoints, making it ideal for scientific study. Though not fine-tuned, it is flexible for various downstream applications like QA. `LLaMA-3.2-1B-Instruct`, by Meta, is an instruction-tuned, multilingual model optimized for tasks like summarization, retrieval, and assistant-style dialogue Grattafiori et al. (2024). It was trained using supervised fine-tuning and Reinforcement Learning from Human Feedback (RLHF), and is designed to produce concise, coherent responses in multi-agent or interactive tasks.

### 4.3 EVALUATION

To evaluate the performance of our models, we use a combination of linguistic and efficiency-related metrics. These metrics help us assess both the quality of the answers generated and the overall system's performance Hu & Zhou (2024). Below is a summary of the metrics used:

**BLEU (Bilingual Evaluation Understudy)** score evaluates the precision of n-gram overlaps between the generated text and the reference text. It includes a brevity penalty to prevent models from

producing overly short outputs. In our case, BLEU quantifies the extent to which the model's predictions match the reference outputs in terms of n-grams. A higher BLEU score indicates that the generated text is closer to the reference in terms of the number of matching n-grams. The BLEU score is computed as:

$$\text{BLEU} = \text{BP} \cdot \exp\left(\sum_{n=1}^{N} w_n \log p_n\right) \tag{1}$$

Here, $p_n$ denotes the modified precision for $n$-grams of size $n$, and $w_n$ represents the weight assigned to each $n$-gram, typically defined as $w_n = \frac{1}{N}$. The variable $N$ specifies the maximum $n$-gram size, which is commonly set to 4. Finally, BP refers to the brevity penalty, defined as:

$$\text{BP} = \begin{cases} 1 & \text{if } c > r \\ \exp\left(1 - \frac{r}{c}\right) & \text{if } c \leq r \end{cases} \tag{2}$$

Here, $c$ denotes the length of the candidate translation, while $r$ denotes the length of the reference translation.

**ROUGE (Recall-Oriented Understudy for Gisting Evaluation)** measures the overlap between the generated text and the reference text. This metric is focused on recall, meaning it looks at how many of the reference n-grams are found in the generated text. It includes variants like ROUGE-N for n-gram recall and ROUGE-L for longest common subsequence (LCS)-based evaluation. ROUGE is typically used for evaluating tasks like summarization, where recall of important information is crucial.

ROUGE-1 and ROUGE-2 measure the overlap of unigrams and bigrams, respectively, between the candidate and reference texts.

$$\text{ROUGE-N} = \frac{\sum_{\text{gram}_n \in \text{Ref}} \min(\text{Count}_{\text{cand}}(\text{gram}_n), \text{Count}_{\text{ref}}(\text{gram}_n))}{\sum_{\text{gram}_n \in \text{Ref}} \text{Count}_{\text{ref}}(\text{gram}_n)} \tag{3}$$

Here, ROUGE-1 corresponds to $N = 1$ (unigrams), while ROUGE-2 corresponds to $N = 2$ (bigrams). The terms $\text{Count}_{\text{cand}}$ and $\text{Count}_{\text{ref}}$ denote the counts in the candidate and reference texts, respectively.

ROUGE-L is based on the Longest Common Subsequence (LCS) between the candidate and reference.

$$\text{ROUGE-L} = \frac{(1 + \beta^2) \cdot \text{LCS}_{\text{F}}}{\text{LCS}_{\text{P}} + \beta^2 \cdot \text{LCS}_{\text{R}}} \tag{4}$$

Here, $\text{LCS}_{\text{P}}$ denotes the precision, computed as the ratio of the longest common subsequence $\text{LCS}(X, Y)$ to the length of the candidate sequence $X$. Similarly, $\text{LCS}_{\text{R}}$ denotes the recall, computed as the ratio of $\text{LCS}(X, Y)$ to the length of the reference sequence $Y$. The measure $\text{LCS}_{\text{F}}$ represents the F-score, which is calculated based on $\text{LCS}_{\text{P}}$ and $\text{LCS}_{\text{R}}$. Finally, $\beta$ is the weighting factor used in the F-score calculation, with $\beta = 1$ being the most common choice.

Here, $X$ is the candidate text and $Y$ is the reference text.

**BERTScore** uses contextual embeddings from a model like BERT to measure the semantic similarity between the predicted and reference tokens. It computes the cosine similarity between token embeddings, allowing it to capture more nuanced meanings rather than surface-level word matches. BERTScore provides a precision, recall, and F1 score based on these embeddings. The BERTScore score is computed as:

$$\text{BERTScore}(c, r) = \frac{1}{N} \sum_{i=1}^{N} \max_{j=1}^{M} \frac{\mathbf{c_i} \cdot \mathbf{r_j}}{\|\mathbf{c_i}\| \|\mathbf{r_j}\|} \tag{5}$$

Let $c_i$ denote the embedding of the $i$-th word in the candidate sentence $c$, and $r_j$ denote the embedding of the $j$-th word in the reference sentence $r$. The candidate sentence contains $N$ words, while the reference sentence contains $M$ words. The cosine similarity between the $i$-th word in the

candidate sentence and the $j$-th word in the reference sentence is represented as $\mathbf{c_i} \cdot \mathbf{r_j}$. Furthermore, $\|\mathbf{c_i}\|$ denotes the norm (magnitude) of the embedding of the $i$-th word in the candidate sentence, and $\|\mathbf{r_j}\|$ denotes the norm of the embedding of the $j$-th word in the reference sentence.

**Semantic Similarity** measures how close the meaning of the generated text is to the reference text. It is computed by comparing the sentence embeddings using cosine similarity. This metric ensures that even if the wording of the generated text differs, it should convey the same underlying meaning as the reference. The Semantic Similarity score is computed as:

$$\text{Semantic Similarity}(c, r) = \frac{\mathbf{c} \cdot \mathbf{r}}{\|\mathbf{c}\| \|\mathbf{r}\|} \tag{6}$$

Here, $\mathbf{c}$ denotes the embedding vector of the candidate sentence $c$, and $\mathbf{r}$ denotes the embedding vector of the reference sentence $r$. The dot product between the embeddings of the candidate and reference sentences is represented as $\mathbf{c} \cdot \mathbf{r}$. Furthermore, $\|\mathbf{c}\|$ denotes the norm (magnitude) of the embedding vector for the candidate sentence, while $\|\mathbf{r}\|$ denotes the norm of the embedding vector for the reference sentence.

**Confidence Score** reflects the model's certainty about its predictions. It is calculated as the average probability assigned to each generated token by the model's softmax function. A higher confidence score indicates that the model is more certain about its output Detommaso et al. (2024); Jurayj et al. (2025). The Confidence Score is computed as:

$$\text{Confidence Score}_i = \frac{e^{z_i}}{\sum_{j=1}^{K} e^{z_j}} \tag{7}$$

Here, $z_i$ denotes the raw logit or score for the $i$-th possible prediction in the model's output, and $e^{z_i}$ represents the exponential of the raw score, ensuring positivity so that it can be normalized. The total number of possible outputs or classes is given by $K$ (for example, the vocabulary size in token prediction tasks). The term $\sum_{j=1}^{K} e^{z_j}$ represents the sum of the exponentials of all logits, which is used to normalize the scores. Finally, ConfidenceScore$_i$ denotes the probability assigned to the $i$-th prediction, representing the model's confidence in that particular choice.

**Tokens per Second** measures the speed at which the model generates tokens, specifically how many tokens are produced per second during inference. A higher throughput means the model is faster at generating text.

$$\text{TPS} = \frac{T_{\text{generated}}}{t_{\text{inference}}} \tag{8}$$

Here, $T_{\text{generated}}$ denotes the total number of tokens generated by the LLM, which may include tokens produced in response to a prompt or during free text generation. The variable $t_{\text{inference}}$ represents the total time (in seconds) taken by the model to generate these tokens, including the entire inference process from input token processing to the generation of the final output. Finally, TPS refers to the Tokens Per Second, indicating the number of tokens the LLM can process or generate per second.

**Memory Usage** tracks the maximum amount of GPU memory used during the inference process. This helps evaluate how efficiently the model utilizes available resources. Monitoring memory usage ensures that the model can operate within the hardware constraints.

**Inference Time** refers to the total duration taken by the model to generate a response, from start to finish. This metric is essential for measuring how quickly the model can produce output, which is critical for real-time applications.

$$\text{InferenceTime} = t_{\text{end}} - t_{\text{start}} \tag{9}$$

Here, $t_{\text{start}}$ denotes the timestamp when the model begins processing the input, such as when tokenization starts or the first token is fed into the model. The variable $t_{\text{end}}$ represents the timestamp when the model completes generating the output, for example when the final token is produced and the output is ready. Finally, InferenceTime refers to the total time required for the model to process the input and generate a response, measured as the difference between the end and start times.

| Configuration | Model Set | ROUGE-1 | ROUGE-2 | ROUGE-L | Avg ROUGE |
|---|---|---|---|---|---|
| Base Model | Phi-2 | 0.3017 | 0.0680 | 0.1577 | 0.1758 |
| | Pythia 1B | 0.2477 | 0.0442 | 0.1280 | 0.1399 |
| | LLaMA-3.2 1B Instruct | 0.2886 | 0.0628 | 0.1455 | 0.1656 |
| | **Combo (Base)** | **0.3394** | **0.3370** | **0.3394** | **0.3386** |
| quantized Model | Quant Phi-2 | 0.3003 | 0.0676 | 0.1575 | 0.1752 |
| | Quant Pythia 1B | 0.2464 | 0.0428 | 0.1270 | 0.1387 |
| | Quant LLaMA-3.2 1B Inst. | 0.2895 | 0.0624 | 0.1459 | 0.1659 |
| | **Combo (Quant)** | **0.3396** | **0.3372** | **0.3396** | **0.3388** |

Table 1: ROUGE score comparison for base and quantized models. Combo (Base) model denotes the combined model architecture of the Base Model. Similarly, Combo (Quant) denotes the combined quantized model.

| Configuration | Model Set | BLEU Score | Semantic Similarity | Confidence Score | BERTScore |
|---|---|---|---|---|---|
| Base Model | Phi-2 | 0.0171 | 0.6659 | 0.6295 | 0.8295 |
| | Pythia 1B | 0.0044 | 0.5470 | 0.4321 | 0.8120 |
| | LLaMA-3.2 1B Instruct | 0.0178 | 0.6667 | 0.5901 | 0.8243 |
| | **Combo (Base)** | **0.2108** | **0.5660** | **0.5799** | **0.8656** |
| Quantized Model | Quant Phi-2 | 0.0176 | 0.6645 | 0.6240 | 0.8292 |
| | Quant Pythia 1B | 0.0046 | 0.5437 | 0.4314 | 0.8114 |
| | Quant LLaMA-3.2 1B Inst. | 0.0213 | 0.6639 | 0.5809 | 0.8245 |
| | **Combo (Quant)** | **0.2109** | **0.5625** | **0.5746** | **0.8655** |

Table 2: Comparison of BLEU, semantic similarity, confidence score, and BERTScore between base and quantized models.

| Configuration | Model Set | Memory Consumption (GB) | Tokens per Second | Time Taken (Minutes) |
|---|---|---|---|---|
| Base Model | Phi-2 | 6.6 | 42.0725 | 1:11:27 (4.29s/it) |
| | Pythia 1B | 3.3 | 104.1318 | 35:10 (2.11s/it) |
| | LLaMA-3.2 1B Instruct | 3.9 | 51.8408 | 1:15:40 (4.54s/it) |
| | **Combo (Base)** | **14.8** | **67.2** | **1:19:01 (4.74s/it)** |
| quantized Model | Quant Phi-2 | 4.4 | 17.0744 | 2:52:31 (10.41s/it) |
| | Quant Pythia 1B | 2.09 | 39.0937 | 1:31:22 (5.48s/it) |
| | Quant LLaMA-3.2 1B Inst. | 2.7 | 25.8326 | 2:23:40 (8.62s/it) |
| | **Combo (Quant)** | **9.19** | **26.39** | **3:15:42 (11.74s/it)** |

Table 3: Performance comparison in terms of memory usage, generation speed, and time taken for base vs quantized models.

## 4.4 EXPERIMENT SETUP

All experiments were conducted on NVIDIA A100 GPUs with 40 GB of memory. For edge device evaluation, we utilized the NVIDIA Jetson Orin Nano, equipped with an 8 GB GPU. Both systems operated on Ubuntu 22.04 LTS.

## 5 RESULTS AND DISCUSSION

### 5.1 RESULTS

This section provides answers to the research questions outlined earlier, based on our experimental findings and analysis.

### Q1: IS THE MULTI-AGENT SYSTEM APPROACH UNIVERSALLY BENEFICIAL ACROSS VARIOUS APPLICATIONS OF LLMS, AND WHAT ARE ITS ASSOCIATED ADVANTAGES AND LIMITATIONS?

Multi-agent systems can offer significant advantages, though they may not always be the ideal solution in every scenario. One of their main strengths is the ability to assign specific tasks to separate agents Guo et al. (2024); LangChain AI (2024). This makes the system easier to develop, test, and maintain. When each agent focuses on a particular domain, it becomes easier to manage the overall system, and communication between agents can be controlled more precisely. Another major benefit is that multi-agent setups allow parallel processing, which can significantly reduce the time required to complete tasks. Because each agent is specialized, the final output is often more

accurate, cleaner, and more reliable - (Tables 1 and 2). However, there are also some drawbacks. When deploying multiple agents, especially on edge devices, we may run into limitations related to memory, power consumption, and overall system latency. Running several models at once requires more computational resources, which can make the system slower or harder to scale in environments with limited hardware capabilities. These trade-offs need to be carefully considered when deciding whether a multi-agent approach is appropriate for a given application.

Q2: DO LARGER MODELS CONSISTENTLY OUTPERFORM SMALLER ONES, OR CAN COMBINATIONS OF QUANTIZED MODELS ACHIEVE SUPERIOR PERFORMANCE COMPARED TO INDIVIDUAL LARGE MODELS?

It is not always the case that larger models with more parameters will outperform smaller ones. While high-parameter models may generally perform well on a wide range of tasks, domain-specific performance often depends on the extent of fine-tuning. A smaller model that has been fine-tuned on a specific domain can, in many cases, outperform a larger, more general-purpose model. This observation highlights the importance of task relevance and domain adaptation over mere model size. The same principle also applies in multi-agent settings, where the effectiveness of each agent is influenced not only by its scale but also by how well it is aligned with the specific sub-task or domain it handles.

Q3: WHY DO COMBINATIONS OF QUANTIZED AND BASE MODELS OFTEN YIELD SIMILAR OVERALL PERFORMANCE, AND WHAT FACTORS CONTRIBUTE TO THIS OUTCOME?

When we quantize a model to INT8, we reduce the precision of its weights, but still preserve the most important parts of the model. This is because the key information is retained, even though the weights are compressed. The core structure, training process, and parameters of the model remain unchanged, which is why the overall performance often doesn't change much after quantization. When using tools such as bitsandbytes for quantization, it tries to reduce the model size while preserving as much crucial information as possible. It quantizes the weights in a way that keeps the key features of the model intact, ensuring that important data isn't lost. Moreover, bitsandbytes performs a process called de-quantization, where it adjusts the quantized weights back to a higher precision during inference, minimizing any loss in accuracy. Additionally, in systems that combine multiple models, like in our setup, the final answer often comes from merging the outputs of different models. Even if one model is quantized and provides a less accurate response, another model might capture the context better. The refined answer is a mix of the best parts from both models, so the final result remains strong. This is why performance from the quantized model can be quite similar to that of the base model as shown in Tables 1 and 2.

Q4: WHY IS THERE A REDUCTION IN TOKENS-PER-SECOND THROUGHPUT IN QUANTIZED MODELS, AND WHAT ARE THE POSSIBLE EXPLANATIONS FOR THIS PHENOMENON?

While INT8 quantization significantly reduces memory usage, we observed a drop in tokens-per-second throughput compared to FP16 and BF16 models. This slowdown can be attributed to several factors. First, although INT8 reduces the size of model weights, it often requires extra processing to convert between INT8 and higher-precision formats during inference. These conversions can add noticeable overhead, especially during decoding where operations are sequential. Second, the efficiency of INT8 kernels depends heavily on hardware support and software optimization. On GPUs like the A100, FP16 and BF16 are highly optimized with native support, whereas INT8 operations may not be as well-tuned, particularly when using general-purpose quantization libraries. As a result, the expected speedup from INT8 quantization may not always be realized in practice, as shown in Table 3.

## 5.2 DISCUSSIONS

### 5.2.1 DECLINE IN ROUGE-1 TO ROUGE-L SCORES

ROUGE is a metric that focuses on exact word matches between the generated output and the reference answer. So if the words overlap, the ROUGE score tends to be high even if the overall meaning isn't entirely correct . In our case, since we're dealing with summarization, the goal is to capture the

semantic meaning of the sentence rather than just matching exact words. That's why we see a lower ROUGE score as in Table 1.

### 5.2.2 SUPERIORITY OF BERTSCORE AS AN EVALUATION METRIC

BERTScore is a more effective evaluation metric in tasks like summarization because it focuses on the semantic meaning of the text rather than just exact word matches. It compares the similarity between the embeddings of the generated and reference sentences Zhang et al. (2019). This allows it to capture whether two sentences mean the same thing, even if they use different words. So, BERTScore gives a better sense of how well the generated text captures the meaning of the original, even if the wording is different.This can be seen in Table 2.

### 5.2.3 REDUCED TOKENS-PER-SECOND IN QUANTIZED MODELS

Although quantization reduces model size and memory usage, it doesn't always improve generation speed as seen in Table 3. In fact, we observed a drop in tokens-per-second when switching from FP16/BF16 to INT8. This happens mainly because INT8 models often need to convert data back to higher precision during inference, especially in tasks like autoregressive decoding where each token is generated one at a time. These frequent conversions add extra processing overhead. Additionally, while FP16 and BF16 operations are highly optimized on modern GPUs like the A100, INT8 kernels are not always as efficient or well-supported. As a result , the performance gains expected from quantization can be offset by slower kernel execution and higher memory access cost.

### 5.2.4 HIGH SEMANTIC SIMILARITY IN COMBINED MODELS

The similarity scores of both the individual models and the combiner model are often very close because the summarizer tends to favor the stronger parts of the responses from the two input models . For example, if one model gives an average answer and the other provides a more accurate or context-rich response, the summarizer will lean more toward the better one. As a result, it might include more content from the stronger model. This can sometimes lead to noticeable differences in similarity scores, especially when there's a large gap in quality between the two input answers. This can be observed in Table 2.

## 6 CONCLUSION AND FUTURE WORKS

In this study, we demonstrated that combining multiple LLMs in an ensemble-like architecture can enhance the quality and reliability of generated responses. Our approach utilizes two base models, `Phi-2` and `Pythia-1B`, which independently generate answers, and a third model, `LLaMA-3.2-1B-Instruct`, that synthesizes these outputs into a single, improved response. Additionally, we introduced optimizations to enable deployment on resource-constrained edge devices without significant loss in performance.

Evaluation on 1K real-world questions showed that our ensemble consistently outperforms individual models, highlighting the potential of collaborative LLM systems to deliver more accurate and helpful answers.

For future work, we plan to extend this framework beyond open-source LLMs by incorporating closed-source models and integrating RAG mechanisms. This would allow the system to leverage domain-specific external knowledge and further improve explainability and adaptability. Moreover, we aim to explore dynamic model selection strategies and scaling techniques to enhance efficiency in diverse real-world applications.

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
