# OpenReview forum: "An Approach for Large Language Model Deployment in Distributed Edge Environments"
_ICLR.cc/2026/Conference — Submitted to ICLR 2026_

### Official Review · Reviewer_Skky · 2025-10-20

**Soundness:** 2
**Presentation:** 3
**Contribution:** 2
**Rating:** 2
**Confidence:** 3

**Summary:**

This paper explores deploying large language models (LLMs) in distributed edge environments by using a multi-agent ensemble approach combined with model quantization. Specifically, two smaller LLMs independently generate responses to user queries, and a third LLM synthesizes these responses into a final output. The authors evaluate performance using a 1K-question subset of the Natural Questions dataset and compare full-precision versus INT8-quantized models. Experimental results indicate that combining multiple smaller models improves answer accuracy and stability over using single models, while quantization significantly reduces computational requirements with limited performance degradation, making the approach more suitable for resource-constrained edge devices.

**Strengths:**

1. The paper tackles the important problem of deploying LLMs in resource-constrained edge environments

2. The results suggest that a multi-model ensemble may improve stability and accuracy in edge environments.

3. The paper is well-written.

**Weaknesses:**

1. The approach largely combines existing ideas: LLM ensembling + summarization + quantization. There is no fundamentally new algorithm or architecture.

2. Using a summarizing LLM as the aggregator might lead to biased responses, but the paper does not evaluate factual consistency post-summarization.

3. It is unclear whether improvements scale when more models are used.

**Questions:**

Please see weakness

---

### Official Review · Reviewer_SPfu · 2025-10-26

**Soundness:** 2
**Presentation:** 2
**Contribution:** 1
**Rating:** 2
**Confidence:** 5

**Summary:**

This paper focuses on solving the issue of deploying LLMs on memory and computation constrained edege devices. Due to the limited resources of edge devices, small langauge models can have poor performance. To improve performance, the paper proposes to use multiple small models to generate a diverse set of responses and then use another model to summarize the responses to form the final answer. The proposed method is evaluated on Natural Questions dataset with quantized and non-quantized models. The experimental results show that the proposed apporache achieved the best performance compared to single models.

**Strengths:**

The proposed approahce is inspired by ensemble method and it achieves the best performance among the compared approaches, which is expected as ensemble method usuall lead to better performance.

**Weaknesses:**

1. The proposed approach doesn't seem to resolve the original constraint, which is the edge devices have limited resources such as memory, as it requires more memories to accomodate at least three LLMs, two for response generation, another one for summarization.
2. The novelty of the paper seems below the bar, as it's well known that ensemble method leads to better performance.
3. For experiments, this work utilizes a subset of the Natural Questions dataset. The paper doesn't discuss how this subset is selected and wheter it's representative. To ensure the proposed approach works in other scenarios, it's better to select a diverse set of evaluation sets  in different areas.
4. Another concern is the metrics. Several classic metrics are proposed for evaluation. However, as the paper itself points out that some of those metrics might not be suitable for evaluation. For example, ROUGE might not be a good metric to evalaute if the final answer is of good quality in terms of meaning of the reponse. LLM-as-judge can be used for better evaluation.

**Questions:**

Please see the weakness section.

---

### Official Review · Reviewer_xr1W · 2025-10-30

**Soundness:** 1
**Presentation:** 1
**Contribution:** 1
**Rating:** 2
**Confidence:** 5

**Summary:**

This paper proposes a multi-agent framework for deploying large language models (LLMs) in resource-constrained edge environments, where multiple smaller LLMs collaboratively generate and synthesize answers to improve accuracy and consistency. Specifically, two base models (Pythia-1B and Phi-2) independently produce responses to the same question, and a third model (Llama-3.2-1B Instruct) aggregates these into a unified, context-aware final answer.

**Strengths:**

(1)This paper proposes a multi-agent framework and demonstrate multiple smaller LLMs collaboratively generate and synthesize answers to improve accuracy and consistency.
(2)The authors also investigate the impact of INT8 quantization on both individual and collaborative models, evaluating performance using metrics like BERTScore, BLEU, and ROUGE on 1,000 samples from the Natural Questions dataset.
(3)This paper demonstrates that combining smaller models can outperform single larger models in both accuracy and stability, and showing that quantization significantly improves deployment efficiency on edge devices with only minor performance degradation.

**Weaknesses:**

(1) First, the paper’s background and introduction are insufficient. It fails to discuss numerous existing works that combine large and small models in similar ways—for instance, ensemble approaches involving multiple models [1][2], as well as prior work with ideas closely resembling this paper’s, where a large model processes and refines the outputs of smaller models before producing a final answer [3]. The paper should include comparisons against these relevant baselines.

(2) The methodology is simplistic and lacks experimental analysis comparing it to alternative fusion strategies. The paper states: “Phi-2 and Pythia-1B were used as question-answering agents, while LLaMA-3.2-1B-Instruct acted as the summarization model responsible for combining and refining their responses.” However, it does not justify why this two-stage architecture (two QA models followed by a summarizer) is necessary or advantageous. What specific roles or complementary capabilities do the two initial models bring? Without such clarification or ablation studies, the design appears arbitrary.

(3) Section 4.3 devotes excessive space to explaining basic evaluation metrics like BLEU, ROUGE, and BERTScore, which are well-established in the field; a single sentence would suffice. Moreover, the paper would benefit significantly from additional analytical experiments that demonstrate the advantages of the proposed ensemble mechanism—such as ablation studies, diversity analysis among agent outputs, or qualitative examples showing how the summarization step resolves inconsistencies or improves answer quality.

References:
[1]URG: A unified ranking and generation method for ensembling language models
[2]SARA: Salience-Aware Reinforced Adaptive Decoding for Large Language Models in Abstractive Summarization
[3]Harnessing multiple large language models: A survey on llm ensemble

**Questions:**

N/A

---

### Official Review · Reviewer_MPnZ · 2025-11-01

**Soundness:** 2
**Presentation:** 2
**Contribution:** 1
**Rating:** 2
**Confidence:** 4

**Summary:**

This paper proposes a multi-agent framework to enhance the consistency and reliability of Large Language Models (LLMs) in question-answering tasks. To address issues like output variability and high computational cost, the authors employ two LLMs (Pythia-1B and Phi-2) to generate independent answers to the same question. A third model (Llama-3.2-1B Instruct) then acts as a synthesizer to combine these responses into a single, unified output. The study further investigates the impact of model quantization (specifically INT8) on the performance of both standalone and collaborative models, using metrics like BERTScore, BLEU, and ROUGE on the Natural Questions dataset.

**Strengths:**

- Important Direction: This paper effectively applies a collaborative, multi-agent strategy—reminiscent of ensemble methods—to the domain of LLMs, which is a promising direction for improving model reliability. The goal itself is important.
- Practical Focus: The work addresses two critical, real-world challenges simultaneously: output inconsistency and computational scalability for deployment on resource-constrained edge devices.
- Integration of Techniques: This paper innovatively combines two distinct approaches—multi-agent systems and model quantization—into a single, cohesive research framework to evaluate their combined potential. The experiment results show that it's an effective method.

**Weaknesses:**

I find the overall thesis lacking in novelty, and its overall standardization and completion are relatively limited.

- **Novelty**: The approach of using two models to output results and a third model for evaluation has been mentioned and widely used in many previous works(please refer to [1], [2], etc.). Additionally, the quantitative methods employed are already common knowledge in the field and do not provide new insights.
- **Standardization**: The writing of the entire thesis is somewhat non-standard, including issues with citation formats and the structure of the introduction. The paper devotes significant space to elaborating on existing evaluation metrics, while the methodology section is overly brief, which diminishes the overall standardization and contribution of the thesis.
- **Completion**: The methodology section reads more like specific experimental settings. Although the paper includes some experimental comparisons, the overall analysis, such as comparisons with related work, is insufficient, and the discussion lacks depth.

[1] Large Language Models Can Self-Improve, EMNLP 2023
[2] N-CRITICS: Self-Refinement of Large Language Models with Ensemble of Critics, NeurIPS 2023

**Questions:**

None.

---

### Meta-Review · Area_Chair_e1iq · 2025-12-28

**Summary:**

This paper proposes a multi-agent framework for deploying LLMs in resource-constrained edge environments, where multiple smaller LLMs collaboratively generate and synthesize answers to improve accuracy and consistency, and a third model (Llama-3.2-1B Instruct) aggregates these into a final answer.

The core idea relies on an ensembling + summarization + quantization pipeline, as discussed by the reviewers. The main concerns are novelty, since this approach has been widely used before, along with other issues such as poor writing in the methodology section, lack of related work, etc. Thus, I recommend rejecting this paper.

**Reviewer Concerns:**

No concerns were addressed by the rebuttal.

**Reviewer Scores:**

No reviewer would change the score.

---

### Decision · Program_Chairs · 2026-01-26

Reject